# Limited options for low-global-warming-potential refrigerants

Mark O. McLinden[1], J. Steven Brown[2], Riccardo Brignoli[3], Andrei F. Kazakov[1] & Piotr A. Domanski[3]

Hydrofluorocarbons, currently used as refrigerants in air-conditioning systems, are potent greenhouse gases, and their contribution to climate change is projected to increase. Future use of the hydrofluorocarbons will be phased down and, thus replacement fluids must be found. Here we show that only a few pure fluids possess the combination of chemical, environmental, thermodynamic, and safety properties necessary for a refrigerant and that these fluids are at least slightly flammable. We search for replacements by applying screening criteria to a comprehensive chemical database. For the fluids passing the thermodynamic and environmental screens (critical temperature and global warming potential), we simulate performance in small air-conditioning systems, including optimization of the heat exchangers. We show that the efficiency-versus-capacity trade-off that exists in an ideal analysis disappears when a more realistic system is considered. The maximum efficiency occurs at a relatively high volumetric refrigeration capacity, but there are few fluids in this range.

[1] Applied Chemicals and Materials Division, National Institute of Standards and Technology, 325 Broadway, Mailstop 647, Boulder, Colorado 80305, USA. [2] Department of Mechanical Engineering, The Catholic University of America, 620 Michigan Avenue, NE, Washington D.C., 20064, USA. [3] Energy and Environment Division, National Institute of Standards and Technology, 100 Bureau Drive, Stop 8631, Gaithersburg, Maryland 20899, USA. Correspondence and requests for materials should be addressed to M.M. (email: markm@boulder.nist.gov).

Hydrofluorocarbons (HFCs), which are currently used as refrigerants in air-conditioning (AC) systems, are potent greenhouse gases, with high values of global warming potential (GWP). Although the present contribution of the HFCs to climate change is small, their contribution is projected to rapidly increase under various scenarios[1]. A phase-down of HFCs is mandated in the European Union[2], and at an October 2016 meeting of the parties to Montreal Protocol, a global phase-down was negotiated[3]. Thus replacement fluids must be found.

A refrigerant is the essential working fluid in a vapour-compression refrigeration cycle; it absorbs heat at a relatively low temperature in the evaporator (for example, the cooling coil in an air conditioner) and releases it at a higher temperature in the condenser (for example, the outside coil). HFC refrigerants were commercialized in the 1990s as replacements for the ozone-depleting chlorofluorocarbons and hydrochlorofluorocarbons. The HFCs are now the dominant refrigerants in new refrigeration, AC and heat-pump equipment. In particular, HFCs are used in small AC systems known as unitary systems: self-contained systems comprising a positive-displacement compressor, condenser, evaporator, and associated fans and controls. R-410A (a blend of HFCs) is currently the dominant refrigerant in such systems. R-22 (a hydrochlorofluorocarbon) was most commonly used prior to R-410A, and it is still commonly used in developing countries. (We use the shorthand nomenclature for these compounds specified in ANSI/ASHRAE Standard 34 (ref. 4); ISO Standard 817 (ref. 5) is substantially equivalent.)

A viable low-GWP candidate must possess a number of other attributes[6], including zero (or very low) ozone-depletion potential, chemical stability within the refrigeration system, thermodynamic properties matched to the refrigeration application, low toxicity and other practical considerations, such as compatibility with the materials of construction. Existing safety codes[7] require nonflammable refrigerants for many applications, but that requirement is being reconsidered.

This work presents the results of a comprehensive search for the best single-component, low-GWP replacement fluids. We search for suitable replacement fluids by applying thermo-dynamic and environmental screening criteria to a comprehensive chemical database[8]. The fluids passing these screens are then simulated in an AC system, with the calculated volumetric refrigeration capacity and energy efficiency serving as additional screens. We conclude that only a limited number of fluids possess the combination of chemical, environmental, thermodynamic and safety properties necessary for a refrigerant in small AC systems and that these fluids are at least slightly flammable. We argue that the presented list of refrigerants is essentially exhaustive. Our focus here is on single-component refrigerants (that is, pure fluids). Refrigerant blends are in common use and offer additional possibilities. We do not consider blends explicitly but, for the sake of completeness, do include several fluids that would not be suitable low-GWP fluids in their own right but that might be useful as a blend component. Our findings give certainty as to the options available to the AC industry in their transition away from high-GWP fluids. It is also important for policy makers to understand these limits and trade-offs as they consider phase-down schedules.

## Results
The current work presents the results of a multi-year study to identify alternatives to the HFC refrigerants. We first briefly summarize interim results from our earlier work, as referenced in the following three subsections.

**Optimal thermodynamic parameters**. As a first step[9], we considered the optimal thermodynamic parameters for a refrigerant. By use of evolutionary algorithms, we examined the performance of hypothetical fluids in several idealized refrigeration cycles and applications. The performance metrics were coefficient of performance (COP, unitless), defined as refrigeration effect (that is, removed heat) divided by the work input to the compressor, and the volumetric capacity ($Q_{vol}$, with units of $MJ \cdot m^{-3}$), defined as the refrigeration effect per unit volume of refrigerant vapour entering the compressor. The COP determines the energy efficiency of an AC system, and $Q_{vol}$ has a large influence on the physical size of the equipment, with larger values of $Q_{vol}$ corresponding to more compact systems. We considered hypothetical fluids—as defined by the liquid–vapour critical temperature ($T_{crit}$), critical pressure ($p_{crit}$), heat capacity of the vapour on a molar basis ($C_p^{\circ}$) and other parameters—meaning that we were not limited to known fluids. The critical temperature was the most important parameter and revealed a trade-off between COP and capacity; a high value of $T_{crit}$ resulted in high COP but low $Q_{vol}$, and vice-versa. A high value of critical pressure resulted in both higher COP and $Q_{vol}$. The optimum value of $C_p^{\circ}$ varied with the refrigeration cycle; low values of $C_p^{\circ}$ were optimal for the basic vapour-compression cycle, while higher values of $C_p^{\circ}$ were optimal for more complex cycles. Low values of $C_p^{\circ}$ are associated with simple molecules (for example, organic molecules with one or two carbons), and most current refrigeration and AC systems use the basic cycle.

**Database screening**. Our search relied on screening a comprehensive database of molecules by applying filters representing different refrigerant selection criteria. The search was carried out in the PubChem database—a listing with >60 million chemical structures[4]. A first screening of this database is described by Kazakov et al.[10]; we summarize here a second screening[11]. All current refrigerants are small molecules, and McLinden[12] provides a thermodynamic basis for this. Thus we limited our search to molecules with ≤18 atoms and comprising only the elements C, H, F, Cl, Br, O, N or S. The choice of elements follows the observation by Midgley[13] that only a small portion of the periodic table would form compounds volatile enough to serve as refrigerants. Despite their ability to deplete stratospheric ozone, Cl and Br were included; a molecule which includes Cl or Br might have a negligible ozone-depletion potential and might be acceptable if it had a very short atmospheric lifetime. These restrictions resulted in 184,000 molecules to be considered further.

Further screens for $320\,K < T_{crit} < 420\,K$ and $GWP_{100} < 1,000$ (GWP with a 100-year time horizon) yielded 138 fluids. The PubChem database does not provide these data for the vast majority of the compounds, so they were estimated using methods based solely on molecular structure; these estimations constituted a major effort of this project[10,14,15]. The limits on critical temperature correspond to fluids usable in small AC systems, with an allowance for the uncertainty in the estimated values of $T_{crit}$. Although refrigerants with values of GWP as low as possible are obviously desirable, fluids with $GWP_{100} < 750$ are, for example, permitted under EU regulations in AC systems with $<3\,kg$ of refrigerant[2]. The full list of 138 fluids is given in Supplementary Table 1, which also lists the $T_{crit}$ and $GWP_{100}$ for each fluid.

The next screens were for chemical stability and toxicity. Compounds with generally unstable functional groups were dropped from further consideration. For example, peroxides (compounds with the $-O-O-$ group) are unstable. Ketenes (compounds with the $-C=C=O$ group) are generally very reactive, and three such compounds were dropped. Allenes have the $-C=C=C-$ group and are characterized as 'difficult to

prepare and very reactive'[16]. Compounds with a carbon-carbon triple bond are generally less stable than those with a double bond; for example, fluoroethyne (FC≡CH) is described as 'treacherously explosive in the liquid state'[17]. There are exceptions, however, and trifluoropropyne was retained.

Attempts to automate the screening of toxicity were not successful. We tested the Toxicity Estimation Software Tool of the US EPA[18], but it estimated, for example, a higher lethal dose for perfluoroisobutene ($(CF_3)_2C=CF_2$), an extremely toxic compound, than for R-134a ($CF_3CFH_2$), a molecule with very low toxicity. (Table 1 and Supplementary Table 1 provide the chemical names of the fluids discussed here.) We concluded that this tool was not optimized for the sorts of small, halogenated

compounds of interest here. Fortunately, at this point, the number of compounds was sufficiently small to allow a 'manual' examination of toxicity data. We considered published toxicity data, where available, making use of a variety of sources, including safety standards, compilations of toxic industrial chemicals, regulatory filings and safety data sheets of chemical manufacturers. We also dropped compounds with two specific groups. Molecules that included the $=CF_2$ group were deemed 'not viable candidates' on the basis of Lindley and Noakes[19] who discuss the '$=CF_2$ structural alert' with regards to R-1225zc ($CF_3CH=CF_2$); this is the observation that the $=CF_2$ group has a high reactivity which is often associated with toxic effects. The presence of a $=CF_2$ group does not assure that a molecule is

**Table 1 | COP and volumetric capacity of selected low-GWP fluids and current HFC and HCFC fluids in the basic, liquid-line/suction-line heat exchanger (LL/SL) and economizer (Econ.) cycles.**

| IUPAC name | Structure | ASHRAE designation | GWP$_{100}$ | COP/COP$_{R-410A}$* | | | Q$_{vol}$/Q$_{vol,R-410A}$* | | |
|---|---|---|---|---|---|---|---|---|---|
| | | | | Basic | LL/SL | Econ. | Basic | LL/SL | Econ. |
| *Hydrocarbons and dimethylether* | | | | | | | | | |
| Ethane | $CH_3$-$CH_3$ | R-170 | 6† | ‡ | | | | | |
| Propene (propylene) | $CH_2=CH$-$CH_3$ | R-1270 | 2† | 1.033 | 1.053 | 1.073 | 0.689 | 0.694 | 0.770 |
| Propane | $CH_3$-$CH_2$-$CH_3$ | R-290 | 3† | 1.014 | 1.042 | 1.058 | 0.571 | 0.579 | 0.640 |
| Methoxymethane (dimethylether) | $CH_3$-O-$CH_3$ | R-E170 | 1† | 0.996 | 1.002 | 1.035 | 0.392 | 0.389 | 0.427 |
| Cyclopropane | -$CH_2$-$CH_2$-$CH_2$- | R-C270 | 86 | 1.018 | 1.021 | 1.045 | 0.472 | 0.467 | 0.510 |
| *Fluorinated alkanes (HFCs)* | | | | | | | | | |
| Fluoromethane | $CH_3F$ | R-41 | 116† | ‡ | | | | | |
| Difluoromethane | $CH_2F_2$ | R-32 | 677† | 1.038 | 1.026 | 1.070 | 1.084 | 1.057 | 1.191 |
| Fluoroethane | $CH_2F$-$CH_3$ | R-161 | 4† | 1.026 | 1.031 | 1.062 | 0.601 | 0.594 | 0.658 |
| 1,1-Difluoroethane | $CHF_2$-$CH_3$ | R-152a | 138† | 0.981 | 0.989 | 1.022 | 0.399 | 0.396 | 0.435 |
| 1,1,2,2-Tetrafluoroethane | $CHF_2$-$CHF_2$ | R-134 | 1120† | 0.967 | 0.991 | 1.024 | 0.348 | 0.352 | 0.385 |
| *Fluorinated alkenes (HFOs) and alkynes* | | | | | | | | | |
| Fluoroethene | $CHF=CH_2$ | R-1141 | <1† | 0.968 | 0.977 | 1.014 | 1.346 | 1.336 | 1.547 |
| 1,1,2-Trifluoroethene | $CF_2=CHF$ | R-1123 | 3 | 0.956 | 0.988 | 1.014 | 1.054 | 1.074 | 1.230 |
| 3,3,3-Trifluoroprop-1-yne | $CF_3$-C≡CH | NA | 1.4 | 0.988 | 1.023 | 1.042 | 0.545 | 0.557 | 0.616 |
| 2,3,3,3-Tetrafluoroprop-1-ene | $CH_2=CF$-$CF_3$ | R-1234yf | <1† | 0.954 | 1.006 | 1.020 | 0.414 | 0.431 | 0.474 |
| (E)-1,2-difluoroethene | $CHF=CHF$ | R-1132(E) | 1 | 1.016 | 1.019 | 1.051 | 0.591 | 0.585 | 0.646 |
| 3,3,3-Trifluoroprop-1-ene | $CH_2=CH$-$CF_3$ | R-1243zf | <1† | 0.964 | 0.997 | 1.019 | 0.372 | 0.379 | 0.417 |
| 1,2-Difluoroprop-1-ene§ | $CHF=CF$-$CH_3$ | R-1252ye§ | 2 | 0.973 | 0.996 | 1.021 | 0.355 | 0.358 | 0.392 |
| (E)-1,3,3,3-tetrafluoroprop-1-ene | $CHF=CH$-$CF_3$ | R-1234ze(E) | <1† | 0.939 | 0.977 | 1.004 | 0.320 | 0.329 | 0.360 |
| (Z)-1,2,3,3,3-pentafluoro-prop-1-ene | $CHF=CF$-$CF_3$ | R-1225ye(Z) | <1† | 0.922 | 0.972 | 0.986 | 0.273 | 0.285 | 0.310 |
| 1-Fluoroprop-1-ene§ | $CHF=CH$-$CH_3$ | R-1261ze§ | 1 | 0.975 | 0.983 | 1.018 | 0.353 | 0.351 | 0.385 |
| *Fluorinated oxygenates* | | | | | | | | | |
| Trifluoro(methoxy)methane | $CF_3$-O-$CH_3$ | R-E143a | 523† | 0.957 | 0.992 | 1.017 | 0.366 | 0.374 | 0.411 |
| 2,2,4,5-Tetrafluoro-1,3-dioxole | -O-$CF_2$-O-CF=CF- | NA | 1 | 0.936 | 0.984 | 0.998 | 0.337 | 0.349 | 0.376 |
| *Fluorinated nitrogen and sulfur compounds* | | | | | | | | | |
| N,N,1,1-tetrafluormethaneamine | $CHF_2$-$NF_2$ | NA | 20 | 0.965 | 1.007 | 1.027 | 0.807 | 0.831 | 0.937 |
| Difluoromethanethiol | $CHF_2$-SH | NA | 1 | 1.010 | 1.019 | 1.054 | 0.582 | 0.580 | 0.642 |
| Trifluoromethanethiol | $CF_3$-SH | NA | 1 | 0.977 | 0.997 | 1.026 | 0.418 | 0.421 | 0.464 |
| *Inorganic compounds* | | | | | | | | | |
| Carbon dioxide | $CO_2$ | R-744 | 1.00† | ‡ | | | | | |
| Ammonia | $NH_3$ | R-717 | <1† | 1.055 | 1.028 | 1.080 | 0.746 | 0.721 | 0.791 |
| *Current HFCs and HCFCs* | | | | | | | | | |
| Pentafluoroethane | $CF_3$-$CHF_2$ | R-125 | 3170† | 0.913 | 0.979 | 0.995 | 0.746 | 0.784 | 0.889 |
| R-32/125 (50.0/50.0) | Blend | R-410A | 1924† | 1.000 | 1.012 | 1.049 | 1.000 | 0.997 | 1.130 |
| Chlorodifluoromethane | $CHClF_2$ | R-22 | 1760† | 1.007 | 1.008 | 1.043 | 0.666 | 0.658 | 0.732 |
| 1,1,1,2-Tetrafluoroethane | $CF_3$-$CH_2F$ | R-134a | 1300† | 0.968 | 0.993 | 1.027 | 0.433 | 0.439 | 0.485 |

Values are for the 'optimized' cycle model and are relative to the performance of R-410A in the basic cycle. GWP$_{100}$ are estimated by the method of Kazakov *et al.*[10] unless noted. The fluids are grouped by chemical class and, within classes, listed in the order of increasing critical temperature.
*Values are relative to those for R-410A in the basic cycle; COP$_{R-410A}$ = 5.35 and Q$_{vol,R-410A}$ = 6.93 MJ·m$^{-3}$.
†Literature value from Myhre *et al.*[32] or EU regulation[2].
‡Fluid would be near-critical or supercritical in the condenser and was not simulated.
§This fluid has *cis* (Z) and *trans* (E) isomers; the predicted values of both were the same.

toxic, but we are aware of only one possible counterexample of R-1123, which has an acute toxicity similar to that of the commercialized refrigerant R-1234ze(E)[20]. (The chronic toxicity of R-1123 has not been reported in the public literature.) The absence of a $=CF_2$ group does not, however, imply that a molecule is of low toxicity. Fluids having the –OF group were also dropped. The –OF group is analogous to the –OH group that defines an alcohol. The bond dissociation energy of the O–F bond, however, is less than one-half that of the O–H bond in an alcohol[21], and the fluorine would likely be reactive with water, forming hydrofluoric acid (a highly toxic compound).

**Cycle simulations—ideal cycle.** The screening of the candidate molecules considered their simulated performance in equipment with operating conditions representative of AC[22]. Specifically, we simulated an evaporator temperature of 10 °C and condensing temperature of 40 °C. We considered the basic vapour compression cycle, the cycle with a liquid-line/suction-line heat exchanger (LL/SL-HX), and a two-stage economizer cycle;

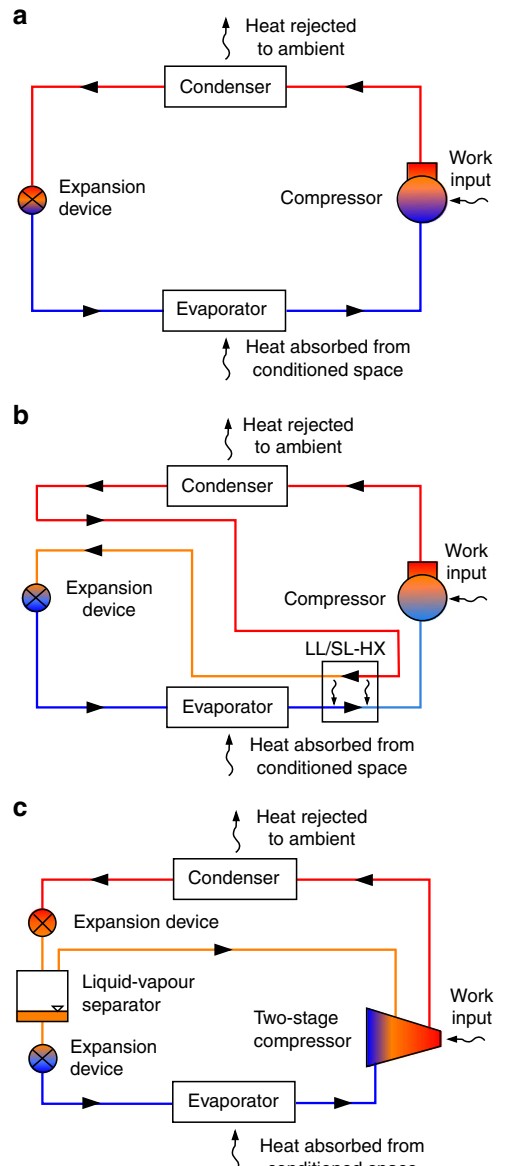

**Figure 1 | Cycles simulated.** (**a**) Basic vapour compression cycle; (**b**) cycle with LL/SL-HX; (**c**) two-stage flash economizer cycle.

these cycles are depicted in Fig. 1. For the representation of refrigerant properties, we used detailed equations of state (EOS) implemented in the NIST REFPROP database[23] where available. However, for a majority of fluids we used the extended corresponding states (ECS) model[24], as discussed in the Methods section.

This screening proceeded in two rounds. The first round of cycle simulations made use of the theoretical CYCLE_D model[25] and provided a first estimate of volumetric capacity and COP[11]. These simulations assumed an ideal cycle with 100% compressor efficiency and no pressure drops. To better elucidate general thermodynamic trends, these simulations were performed on the full list of 138 low-GWP candidates (that is, including those dropped as unstable or toxic) as well as an additional 8 refrigerants in current use. These results are given in Fig. 2 and Supplementary Table 1. This figure clearly shows the COP (efficiency) versus capacity trade-off that results from an ideal analysis based only on thermodynamic properties[26]. At this stage, we dropped fluids with a volumetric capacity less than one-third that of R-410A or a COP < 5. (For R-410A in the ideal cycle, $Q_{vol} = 6.62\ MJ \cdot m^{-3}$ and COP = 7.41. The volumetric capacity of R-22 is 66% that of R-410A, so this would correspond to dropping fluids with a capacity less than one-half that of R-22.) The stability, toxicity and performance screens yielded a set of 27 low-GWP fluids that were then simulated in greater detail.

**Cycle simulations—optimized cycle.** The second round of simulations made use of a more advanced 'optimized' cycle model that provided a more realistic representation of an air conditioner employing typical forced-convection, air-to-refrigerant heat exchangers, which were optimized for a particular refrigerant. In this type of heat exchanger, the refrigerant undergoes a phase change as it flows down the inside of a tube and exchanges heat with air on the outside of the tube. Specifically, the new model accounted for the effect of optimized refrigerant mass flux, which enhances the refrigerant heat transfer coefficient at an acceptable penalty of the pressure drop, as described by Brown et al.[27] The simulation model maintained the same heat flux in the evaporator through all simulations, which is a prerequisite for a fair rating of competing refrigerants[28]. The isentropic efficiency of the compressor was a function of the refrigerant properties and averaged 70%. Here the relative ranking of fluids differs from a ranking based only on thermodynamic properties; it is, however, more representative of a fluid's performance in an AC system in commercial production, which would be optimized for the refrigerant being used.

This screening process yielded 27 low-GWP fluids we deem to be the best single-component low-GWP replacements for unitary AC systems, see Table 1. This list is a subset of the 138 candidates, with the deletion of those that have low $Q_{vol}$, low COP or are unstable or toxic. We also included in Table 1 four currently used refrigerants, as well as carbon dioxide, ethane, R-41 and R-1225ye(Z), as discussed in the Methods section. Refrigerant blends are currently in common use, and the fluids in Table 1 also constitute the components of future blends. The list includes a small number of novel molecules that have not been previously considered as refrigerants (at least publicly), but a majority of the fluids are well known, including ammonia (R-717) and propane (R-290), or are the focus of current research in the refrigeration industry, that is, the fluorinated alkenes (also known as hydrofluoroolefins or HFOs). The HFOs constitute the largest group in the list with nine fluids. The other fluids are halogenated alkanes, halogenated oxygenates, hydrocarbons, halogenated nitrogen and sulfur compounds and inorganic compounds.

The COP and $Q_{vol}$ of the candidate fluids, based on the optimized model, are presented in Table 1 and Fig. 3. The COP

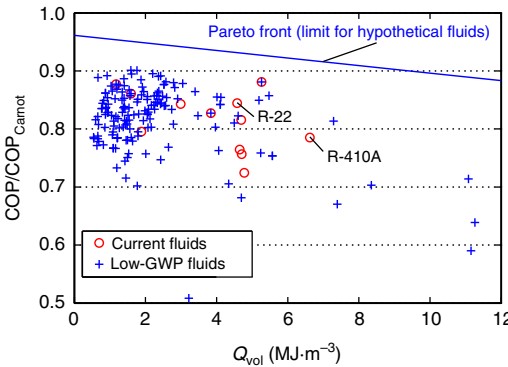

**Figure 2 | Results of ideal cycle analysis for low-global-warming-potential and current fluids.** This analysis shows the trade-off between the coefficient of performance (COP) and volumetric capacity. The majority of low-GWP fluids have low volumetric capacity relative to that of R-410A. The 'Pareto front' line shows the thermodynamic limit of performance for fluids in the ideal vapour compression cycle, as discussed by Domanski et al.[9].

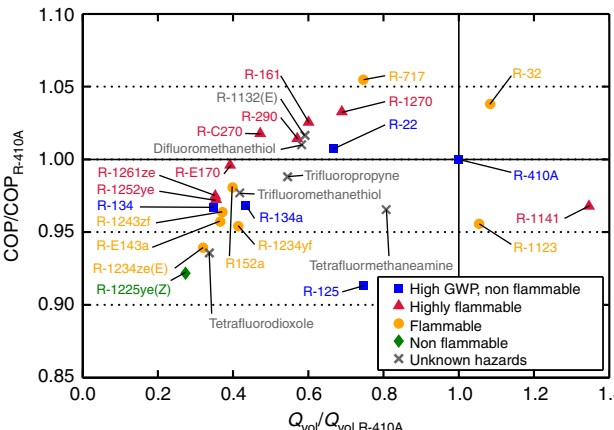

**Figure 3 | Coefficient of performance and volumetric capacity of selected low-global-warming-potential fluids.** Results are shown for the basic vapour compression cycle; values are relative to those for R-410A and are calculated with the 'optimized' cycle model.

ranged from $-7.8\%$ to $+5.5\%$ relative to that of R-410A in the basic vapour compression cycle. Ammonia showed the highest COP, better than that for R-410A by 5.5%. Beyond ammonia, which is toxic, mildly flammable and presents materials compatibility issues, the COPs of R-32, propene (R-1270), R-161, R-1132(E), propane (R-290), cyclopropane (R-C270) and difluoromethanethiol are also above the R-410A baseline. The COPs of the remaining fluids are lower. Mildly flammable R-32 has a COP and $Q_{vol}$ higher than that of R-410A, but this advantage comes with a $GWP_{100}$ of 677. R-134 and R-E143a have $GWP_{100}$ values of 1120 and 523, respectively. Three fluids have GWPs within the 80–150 range, and GWPs for the remaining fluids do not exceed 20. Except for R-32, R-1123 and R-1141, the listed fluids have $Q_{vol}$ lower than R-410A and would thus require a larger compressor—by at least 25%—and, for a majority of the candidates, more than twice as large—to provide the same capacity as R-410A. Table 1 does not provide COP and $Q_{vol}$ for carbon dioxide, ethane and R-41 because their $T_{crit}$ are low and they may require a different (that is, transcritical) cycle, depending on operating conditions. We list them because they may be suitable as a component of a blend. In general, fluids with a low $T_{crit}$ (corresponding to high $Q_{vol}$) suffer performance degradation at high ambient temperatures.

Our assumed conditions were an evaporator temperature of $10\,°C$ and condensing temperature of $40\,°C$; these are representative for the majority of AC systems in North America, Europe and Japan. The conclusions regarding suitable fluids for higher-ambient temperatures may differ, however, as exemplified by the concerns regarding the viability of current refrigerants having relatively low critical temperatures, such as R-410A ($T_{crit} = 344.5\,K$) and R-32 ($T_{crit} = 351.3\,K$), in these applications. Consequently, other refrigerants in Table 1 with low critical temperatures, that is, R-1141 ($T_{crit} = 327.1\,K$), R-1123 ($T_{crit} = 343.0\,K$) and tetrafluoromethaneamine ($T_{crit} = 342\,K$), are probably not viable candidates for systems operating in high-ambient-temperature conditions.

Compared with R-410A, the COP of R-22 is slightly higher, and the $Q_{vol}$ is lower; thus the values in Fig. 3 and Table 1 can be referenced to an R-22 baseline by multiplying the COP and $Q_{vol}$ values by 0.993 and 1.502, respectively. The conclusions about alternatives to R-22 are the same, except that three additional fluids (R-717 (ammonia), R-1270 (propene) and tetrafluoromethaneamine) have slightly higher, rather than lower, values of $Q_{vol}$ when referenced to R-22.

The results for the LL/SL-HX and economizer cycles are qualitatively similar to the basic cycle and are listed in Table 1 and depicted in Fig. 4. Similar to the basic cycle (Fig. 3), the best COP values correspond to $Q_{vol}$ values that are at least 60% of that for R-410A; the upper range of optimal $Q_{vol}$ is somewhat extended above 110% for the economizer cycle because we normalized the data with R-410A values for the basic cycle. The LL/SL-HX cycle (Fig. 4a) provides a performance benefit to fluids with a high molar heat capacity and degrades the performance of fluids with a small molar heat capacity (which are best performers in the basic cycle). Consequently, the spread of COP values in Fig. 4a is smaller than that shown in Fig. 3. The economizer cycle (Fig. 4b) increases the COP for all refrigerants, although the increase is larger for the fluids having a high molar heat capacity.

## Discussion

Unlike the COP versus $Q_{vol}$ trade-off observed for the ideal analysis (Fig. 2), the results of the optimized cycle simulations (Figs 3 and 4) show a maximum in COP corresponding to $Q_{vol}$ of approximately 60–110% that of R-410A. Relative to fluids with low values of $Q_{vol}$, the high-$Q_{vol}$ fluids have lower values of $T_{crit}$ and operate at higher pressures; the result is that the cycle operates near the critical point and suffers increased irreversibilities in the expansion process. This effect applies to both the ideal and more detailed analyses. However, the ideal analysis neglects the fact that the pressure drop in the heat exchangers (condenser and evaporator) extracts a smaller COP penalty on the high $Q_{vol}$ (that is, high pressure) fluids when the heat exchangers are optimized. (The benefit of this optimization will also be affected by the relative heat transfer resistance on the refrigerant side and air side of the heat exchangers, as discussed in the Methods section.) An additional effect is that the low-$Q_{vol}$ fluids tend to be more complex molecules (see Supplementary Table 1). For example, R-32 (one of the best fluids in Fig. 3) is based on a single carbon atom, and R-410A is a blend of the single-carbon R-32 and two-carbon R-125. In contrast most of the fluids with $Q_{vol} < 0.4 \cdot Q_{vol,R-410A}$ are three-carbon compounds; greater complexity is associated with higher values of viscosity, which would increase the pressure drop and lower the COP.

This preference for high-pressure fluids does not apply to all types of systems. For example, large central-plant chillers typically employ shell-and-tube heat exchangers that have very-low refrigerant-side pressure drops; they often employ low-pressure

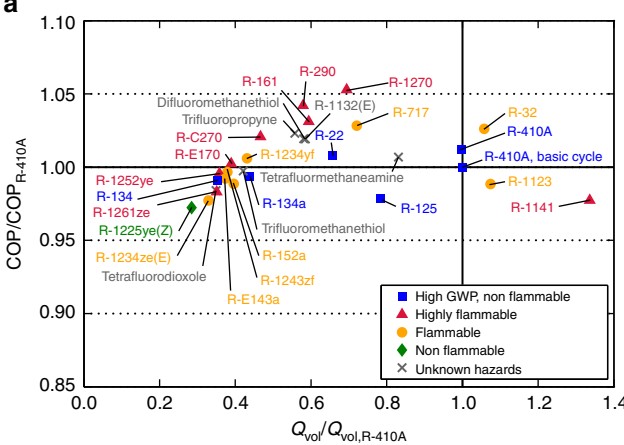

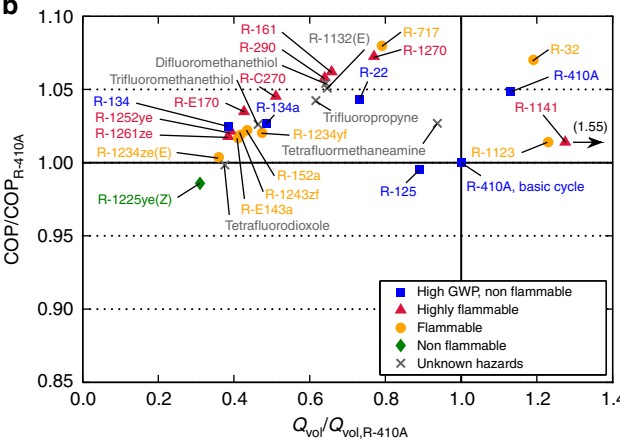

**Figure 4 | Optimized coefficient of performance versus volumetric capacity for the liquid-line/suction-line heat exchanger and economizer cycles.** Values plotted are relative to those for R-410A in the basic cycle ($COP_{R-410A} = 5.35$ and $Q_{vol,R-410A} = 6.93\,MJ \cdot m^{-3}$); (**a**) cycle with LL/SL-HX; (**b**) two-stage flash economizer cycle.

refrigerants and achieve high COPs. The complexity of such equipment is, however, impractical for small AC systems. Very small systems, such as home refrigerators, typically use medium-to-low-pressure refrigerants (for example, R-134a or isobutane) because the use of a high-pressure fluid, such as R-410A, would result in such a small compressor that mechanical losses would become large.

Most of the candidate fluids identified are flammable or mildly flammable. The refrigerant classification standards[4,5] assign a fluid to one of four flammability groups based on their lower flammability limit, heat of combustion and burning velocity. Ranging from nonflammable to most flammable, these are (in order) 1, 2L, 2, and 3. Classes 2L and 2 are denoted by yellow symbols in Figs 3 and 4. Most current AC systems use class 1 (nonflammable) fluids, although the 'mildly flammable' class 2L fluids (those with a burning velocity $< 10\,cm \cdot s^{-1}$) are being considered for use. R-1225ye(Z) is nonflammable but has a capacity of only about one-fourth that of R-410A. R-1225ye(Z) has low acute toxicity[19], but it exhibited toxic effects upon longer exposures at the relatively low levels of 500 to 1,000 p.p.m. (ref. 29). Carbon dioxide is also nonflammable (class 1), but it operates at very high pressures in a transcritical cycle owing to its low critical temperature. Because of this, the use of $CO_2$ would require extensive redesign of current equipment types and was not simulated here.

We identified six novel molecules in the screening: tetrafluorodioxole, trifluoromethanethiol, trifluoropropyne, difluoromethanethiol, (E)-1,2-difluoroethene (R-1132(E)), and tetrafluoromethaneamine. Few data could be found on these fluids, and they present unknown risks. None of them are particularly compelling from a performance standpoint. Difluoromethanethiol and (E)-1,2-difluoroethene, for example, have predicted COPs slightly higher than R-410A but $Q_{vol}$ values that are about 40% lower than that of R-410A. They are flammable (in addition to possible other hazards), and their COP and $Q_{vol}$ are very similar to propane (R-290). This raises the question, why take on the unknown risks of these fluids when one could use the somewhat more flammable but well-known propane? R-134 has a relatively high $GWP_{100}$ of 1120. It was investigated in the 1990s but never used commercially; it may be of interest as a blend component.

Our major conclusion is that the viable candidates for single-component low-GWP alternatives for small AC systems are very limited, especially for refrigerants with volumetric capacities similar to R-410A. Fluids with good COP and low toxicity are available, but all are at least slightly flammable. Nonflammable candidates exist among the fluids with low volumetric capacity, but use of such fluids in small AC systems would require extensive redesign and may result in lower COP. Blends offer additional possibilities, and the refrigeration industry is actively investigating blends of HFCs and HFOs with the intent of reducing or eliminating flammability with the trade-off of increased GWP. Although our study focussed on unitary AC systems (that is, residential and small commercial single-package and split systems), the general conclusions would apply also to room AC units and to refrigeration and heat-pumping systems currently using R-410A or R-22. The list of suitable fluids for systems operating at high ambient conditions would be reduced (as presented in the Results section).

Accepting thermodynamic arguments that viable refrigerants are restricted to small molecules, there are a finite number of ways to combine the selected elements into stable molecules. It is our contention that the presented screening process has yielded a list (see Table 1) of the 'best' low-GWP fluids allowed by chemistry, that is, it is highly unlikely that any better-performing fluids will be found, and unknown risks associated with the lesser-known fluids may further reduce the list.

Refrigerant blends offer a compromise between flammability and GWP: for example, a low-GWP but flammable fluid blended with a nonflammable but high-GWP fluid could result in a nonflammable fluid with a moderate value of GWP (order of 500) or a slightly flammable refrigerant with a low GWP (order of 150). The list presented in Table 1 (with the addition of the 'current' HFCs listed) also encompasses the fluids that would serve as blend components. The refrigeration industry has been very actively developing such blends, and in parallel, significant work has been undertaken in support of refrigerant safety standards that could allow the safe use of flammable refrigerants in specific applications. The results of the present study have confirmed these courses of action.

Our findings give certainty as to the options available to the AC industry in their transition away from high-GWP fluids. It is also important for policy makers to understand these limits and trade-offs as they consider phase-down schedules.

## Methods

**Database screening.** The PubChem database[8] is a listing of 157 million chemical substances and 60 million unique chemical structures, and we considered it as exhaustive for the small molecules that would be viable as refrigerants. The entries in PubChem are provided by a wide variety of contributors, including chemical vendors, university laboratories and government agencies, including the US

Environmental Protection Agency. More than 16 million chemical structures were obtained from six million US, European and World Intellectual Property Organization patent documents. In addition to commercially available chemicals, it includes compounds that have been synthesized only on a laboratory scale and also entirely hypothetical molecules.

We carried out two screenings of this database. The first screening is described by Kazakov et al.[10], who generated a subset of 56,203 molecules of ≤15 atoms and comprising only the elements C, H, F, Cl, Br, O, N or S. Also excluded at this stage were ions, radicals and molecules enriched in specific atomic isotopes. Kazakov et al.[10] developed a method for predicting the $GWP_{100}$ of compounds based only on their molecular structure; the method combined estimates of the radiative efficiency and atmospheric reactivity with the hydroxyl radical. The trade-off of computational efficiency versus accuracy was a major consideration given the large number of compounds to screen. The method achieved a logarithmic root-mean-square deviation of 3.0 for the $GWP_{100}$ estimate compared with available literature values, and this accuracy was adequate for screening purposes. Applying a screen of $GWP_{100} < 200$ resulted in 52,565 compounds, that is, the vast majority of these molecules have short atmospheric lifetimes and low GWP. Further screens for toxicity (based on the list of markers and associated rules of Lagorce et al.[30]) and flammability (based on a new estimation method[10]) reduced the list further to 20,277. The critical temperature of these were estimated using the method of Kazakov et al.[14], and a constraint of $300 \text{ K} < T_{crit} < 550 \text{ K}$ was applied. The number of candidates was further reduced to 1,234 by eliminating unstable or highly reactive functional groups.

The list of 1,234 compounds was examined by McLinden et al.[31] and compared against the optimum thermodynamic parameters for a refrigerant that were revealed by the analysis of Domanski et al.[9] The main thermodynamic criterion was critical temperature and only 62 fluids had $300 \text{ K} < T_{crit} < 400 \text{ K}$. (The higher upper limit on $T_{crit}$ in the database screening was selected before the results of Domanski et al.[9] were obtained and also to encompass high-temperature applications.) The 62 fluids were examined with regards to thermodynamic parameters and a more detailed consideration of toxicity and chemical stability. Only a dozen fluids remained as viable candidates.

Given this small number of viable candidates, a second screening of the PubChem database was carried out (McLinden et al.[11]). In this screening, the maximum size of the molecule was increased to 18 atoms (with the intent of enabling a future search for fluids suitable for high-temperature applications); the result was 184,000 compounds. The most significant changes in the second screening, however, were the elimination of the screens on toxicity and flammability, and a relaxing of the GWP screen to $GWP_{100} < 1,000$. These were carried out to avoid prematurely excluding promising candidates and in the recognition that a flammable fluid or one with a moderate value of GWP might be acceptable in some applications or as a component of a blend. The calculation of critical temperature was carried out as the first filter using the method of Kazakov et al.[14]; 1,728 low-GWP compounds had $T_{crit} < 550 \text{ K}$ and 138 candidates had $T_{crit} < 420 \text{ K}$.

Cycle simulations (as described below) were carried out on these 138 fluids. A literature search on the toxicity and chemical stability of the candidates was conducted, except that fluids with a low volumetric capacity in the ideal vapour compression cycle were not considered further. The selection of a fluid for inclusion in the 'final' list (that is, Table 1) was based on (1) $Q_{vol} > 0.33 \cdot Q_{vol,R410A}$ (that is, $Q_{vol} > 2.2 \text{ MJ} \cdot \text{m}^{-3}$); (2) COP > 5; (3) low toxicity (or, at least, no documentation of high toxicity); and (4) acceptable chemical stability. Several additional fluids were also included in Table 1 to present a 'complete' list of options for unitary AC applications: Carbon dioxide is the only low-toxicity, nonflammable, high-pressure fluid that was identified; it would operate in a transcritical cycle and was not simulated here. Ethane and R-41 (fluoromethane) would also operate near their critical temperature in an AC application; they would be unlikely single-component refrigerants, but they might be useful as blend components. R-134 (1,1,2,2-tetrafluoroethane) is a HFC with $GWP_{100} = 1,120$ (ref. 32); it is nonflammable and, despite its relatively high value of GWP, it might be useful as a blend component. We included R-1123 and R-1225ye(Z) in Table 1 and Fig. 3 because these fluids are (or were at one time) the subjects of active commercialization efforts.

Propyne (CH$_3$-C≡CH) and propa-1,2-diene (CH$_2$=C=CH$_2$) present an interesting situation. These fluids have normal boiling points of $-23$ and $-34 \,^{\circ}\text{C}$, respectively, and moderate values of $Q_{vol}$ in the ideal-cycle simulation (see Supplementary Table 1). The two form an equilibrium mixture, which is stable and of low toxicity[33]. The composition of the mixture, however, depends on temperature. Thus neither could be considered a 'single-component refrigerant.' For this reason, we did not simulate them in the 'optimized' cycle model (described below) nor included them in Table 1. Nevertheless, the mixture does represent a further possibility in applications where a flammable refrigerant is acceptable.

**Properties estimation.** For the cycle simulations carried out on the list of 138 fluids, we employed the detailed EOS models for the thermodynamic properties implemented in the NIST REFPROP database[23], where available. Such EOS were available for only 22 fluids. For the majority of the fluids, we used the ECS model of Huber and Ely[34]. The ECS model provides a good and thermodynamically consistent representation of the thermodynamic properties of a fluid given only the

critical temperature, critical pressure, acentric factor $\omega$ (a parameter related to the slope of the vapour pressure with respect to temperature) and the heat capacity of the vapour. Although the screening of the PubChem database made use of the critical temperatures estimated by the method of Kazakov et al.[14], the $T_{crit}$, $p_{crit}$ and $\omega$ used in the ECS model were estimated with the more recent method of Carande et al.[15] based only on the molecular structure. The vapour heat capacity was estimated by a statistical–mechanical model described by Kazakov et al.[10] The ECS model has additional fitting parameters, but these were shown by Domanski et al.[9] to have a small effect on the cycle performance and were set to zero. McLinden et al.[24] provide more details on calculating refrigerant properties with the ECS model.

For six fluids, we had the intermediate case of very limited experimental data, such as normal boiling point temperature, and for these, we applied traditional group contribution methods, which require boiling-point data, to estimate the critical parameters and $\omega$. We implemented four such methods: Ambrose (as described by Reid et al.[35]), and the methods of Joback, Marrero and Wilson (as described by Poling et al.[36]).

The heat exchanger optimizations carried out for the final set of fluids listed in Table 1 required, in addition to the thermodynamic properties, the transport properties of thermal conductivity and viscosity. These were calculated with the models implemented in the NIST REFPROP database, where available. For the remaining fluids, we used a dilute-gas model based on the Lennard–Jones (L–J) fluid, with the L–J parameters estimated with the method of Chung[37], combined with the ECS model of Huber et al.[38] for the dense-gas contribution. We used the model of Chae et al.[39] for estimating the surface tension.

**Cycle simulations.** The basic vapour-compression refrigeration cycle and two modifications of this cycle were considered; these are shown in Fig. 1. In the 'basic' vapour-compression refrigeration cycle, Fig. 1a, refrigerant vapour enters the compressor and is compressed to a relatively high pressure, such that its saturation temperature is above that of the high-temperature heat sink (for example, outside air for an AC). The hot refrigerant vapour condenses to a liquid in the condenser, releasing the heat of vapourization at an approximately constant temperature. In most AC systems, refrigerant flows inside the tubes of the condenser, which are cooled by air blown across fins on the outside of the tubes. The liquid refrigerant then flows through an expansion device, which lowers the pressure in an adiabatic (constant enthalpy) process. The expansion device can be as simple as a long capillary tube, but it is usually a thermostatically or electronically controlled valve in most AC equipment. In the expansion process, a portion of the liquid flashes to vapour, cooling the refrigerant to the saturation temperature corresponding to the pressure in the evaporator. The remaining liquid vapourizes in the evaporator (which is typically also a finned, air-to-refrigerant heat exchanger), extracting heat from the cooled space (for example, indoor air). The refrigerant vapour then flows to the compressor, completing the cycle.

The LL/SL-HX (Fig. 1b) adds an additional internal heat exchanger to the basic cycle. The hot liquid refrigerant exiting the condenser is cooled by rejecting heat to the cold refrigerant vapour exiting the evaporator. This reduces the quantity of refrigerant that flashes to vapour upon exiting the expansion device; this increases the refrigeration effect in the evaporator, which increases the COP. Simultaneously, however, it heats the vapour entering the compressor, which increases the compression work—an effect which decreases the COP. As a result, this internal heat exchanger can increase or decrease the COP of the cycle, depending on the properties of the fluid; those with low values of the vapour heat capacity on a molar basis (corresponding to small molecules, such as ammonia) suffer a COP penalty, while relatively complex molecules (such as those with three carbons) usually benefit[40].

The final cycle considered here is the two-stage flash economizer cycle, shown in Fig. 1c. Here there are two expansion devices, with a liquid–vapour separator between them. After the first expansion device, the refrigerant that has flashed to vapour is sent directly to the compressor, rather than flowing through the evaporator. As this portion of the refrigerant flow is already vapourized, it would not contribute any refrigeration effect in the evaporator; it is at a pressure intermediate between that in the condenser and evaporator and thus requires less compression work to raise it back to the condenser pressure. The economizer cycle yields a higher COP than the basic cycle for all fluids, but this comes at the expense of additional components and a more complex compressor.

The 'optimized' cycle model used to simulate the results presented in Table 1 was derived from the CYCLE11 model of Domanski and McLinden[28] with the addition of an optimization of the refrigerant circuitry in the evaporator and condenser to maximize the COP. The model represents the heat duties, $Q$, of the evaporator and condenser (in a cross-flow configuration) through the overall heat transfer coefficient, $U$, heat transfer area, $A$, and mean effective temperature differences, $\Delta T$, where the heat transfer rates are given by $Q = UA \cdot \Delta T$. The mean effective temperature differences, $\Delta T$, are determined from the temperature profiles of the heat sink and heat source and the refrigerant-side temperature profiles in the evaporator and condenser. The solution scheme divides the heat exchangers into as many as 128 segments and includes sections in the condenser for vapour desuperheating, two-phase condensation and liquid subcooling and sections in the evaporator for two-phase evaporation and vapour superheating. The 'optimized' model is not a detailed simulation model. Rather it accounts for the effects of

refrigerant thermophysical properties on heat transfer coefficients and refrigerant pressure drops in the heat exchangers in relation to those of a reference refrigerant in a reference system (either simulated with a much more detailed model or experimentally measured), and it assumes heat transfer resistances on the air sides of the heat exchangers to be constant, as presented by Brown et al.[27,41]

The refrigerant flows through multiple tubes in the evaporator and condenser, and the model optimizes the COP by distributing the flow among one or more refrigerant circuits in each heat exchanger. An increase in the number of tubes per refrigerant circuit in effect lowers the refrigerant mass flux per tube leading to a lower refrigerant-side heat transfer coefficient more or less proportionally with mass flux while at the same time lowering the refrigerant-side pressure drop per tube more or less quadratically with mass flux. Conversely, decreasing the number of tubes per refrigerant circuit has the opposite effects on the refrigerant-side heat transfer coefficient and pressure drop. These effects result in an optimum refrigerant-side refrigerant flow path that maximizes COP by striking a balance between the positive influence on COP of higher refrigerant mass flux through increasing heat transfer coefficient and the negative influence on COP of higher refrigerant mass flux through increasing pressure drop.

**Uncertainties.** The screening of candidate molecules in this study was based primarily on values of $GWP_{100}$, the thermodynamic properties and the simulated cycle performance in an AC application (which, in turn, depended on the thermodynamic properties). The $GWP_{100}$ for 17 of the 27 fluids presented in Table 1 are available in the literature, and for the other 10 (and a large majority of the fluids in the full list of 138 fluids in Supplementary Table 1) it was estimated by the method of Kazakov et al.[10] Based on 95 fluids with literature values of $GWP_{100}$, Kazakov et al.[10] determined that the logarithmic root-mean-square deviation of the estimated $GWP_{100}$ corresponded to a factor of 3.0. Although this is a large uncertainty, it is adequate for screening purposes, especially considering that a $GWP_{100}$ of the order of 1.0 was estimated for a large fraction of the compounds considered—a value of 0.3 or 3.0 would still put such a fluid in the category of 'extremely low GWP.'

The uncertainties in the simulated cycle performance arise from two sources. The first source stems from the assumptions and idealizations made in the cycle model. Here all fluids were simulated with the same assumptions, and we were concerned only with relative differences between fluids.

The second (and major) source of uncertainty in the simulation results stemmed from the thermodynamic properties. For the 22 fluids, we made use of the comprehensive, high-accuracy EOS implemented in the NIST REFPROP database[23]; for these fluids, we consider the uncertainties in the thermodynamic properties to be negligible for the purposes of the present study. For the remaining fluids, the properties were calculated with the ECS model[24,34], based on estimated values of $T_{crit}$, $p_{crit}$, $\omega$ and ideal-gas heat capacity ($C_p^{\circ}$). Uncertainties arise from the estimates of $T_{crit}$, $p_{crit}$, $\omega$ and $C_p^{\circ}$. The standard uncertainties of the group contribution estimates based on normal boiling point (a method used for six fluids) were estimated by Brown et al.[42] to be 1.0, 10, 12 and 6.5% for $T_{crit}$, $p_{crit}$, $\omega$ and $C_p^{\circ}$, respectively. To estimate the uncertainties in cycle performance, we simulated three of the fluids (R1132(E), R1141 and R1225ye(Z)) with varying estimates of the input parameters. A variation in $T_{crit}$ of $\pm 2\%$ resulted in uncertainties in COP of $-4.0\%/+3.1\%$ and uncertainties in $Q_{vol}$ of $-15.1\%/+17.4\%$. A variation in $p_{crit}$ of $\pm 10\%$ resulted in virtually no change in COP and uncertainties in $Q_{vol}$ of $+10.0\%/-10.0\%$. A variation in $\omega$ of $\pm 15\%$ resulted in virtually no change in both COP and $Q_{vol}$. Finally, varying $C_p^{\circ}$ by $\pm 15\%$ resulted in uncertainties in COP of $-2.1\%/+1.9\%$ and uncertainties in $Q_{vol}$ of $-3.6\%/+3.6\%$. These simulations were for the basic cycle and we assumed similar uncertainties for the LL/SL-HX and economizer cycles.

The sensitivity study described above for R1132(E), R1141 and R1225ye(Z) will apply for all the fluids. Additional fluids were estimated by the method of Carande et al.[15], who provide 'median average deviations' (MAD) for the estimated properties; the MAD approximates a standard uncertainty. The cycle performance simulations are most sensitive to uncertainties in the critical temperature, and the MAD for the $T_{crit}$ estimated by the method of Carande et al.[15] averaged 16.5 K (4.3%) or substantially larger than the group contribution method based on normal boiling point.

Although the uncertainties in COP and $Q_{vol}$ are relatively large for individual fluids, the overall trends and conclusions remain valid. The minimum value of $Q_{vol}$ for inclusion in Table 1, for example, was 33% that of the R-410A baseline—an allowance that was selected in view of the uncertainty in the cycle performance. Likewise, the maximum value of $T_{crit}$ for inclusion in the evaluation (that is, Supplementary Table 1) was 420 K compared with $T_{crit} = 345$ K for R-410A. It is highly unlikely that a fluid that would be a close replacement for R-410A was passed over because of uncertainties in the method.

**Data availability.** All of the data depicted in Figs 2–4 and discussed in the text are given in Table 1 or Supplementary Table 1. These data were calculated with NIST databases, cited literature sources and other commercial and open-source tools.

The ideal-cycle simulations (Fig. 2 and Supplementary Table 1) made use of the NIST CYCLE_D cycle-simulation model[25], available from the NIST Standard Reference Data Program; see: http://www.nist.gov/srd/nist23.cfm. The new 'optimized' cycle model (Figs 3 and 4 and Table 1) will be documented in a

forthcoming paper; a preliminary version is available upon request. For both models, the thermophysical properties were calculated with the NIST REFPROP database[23], see http://www.nist.gov/srd/nist23.cfm. For the 22 fluids, the standard property formulations in REFPROP were used. For the remaining fluids, ad hoc fluid data files were generated with the ECS method, as outlined in the main text and detailed in ref. 24 or (for the fluids with limited experimental data) with the estimation methods[35,36] described in the main text.

The $GWP_{100}$ values and some of the property estimates were based on quantum mechanical calculations; specifically, the vibrational frequencies, infrared intensities, radiative efficiency, ideal-gas heat capacities and enthalpies of formation were computed at the PM6 level of theory with Gaussian 09 Rev B.01 (ref. 43).

The critical properties estimated by the method of Kazakov et al.[14] made use of MOPAC version 6 (ref. 44), CODESSA version 2.7.9 (ref. 45) and LIBSVM version 3 (ref. 46). The critical properties estimated by the method of Carande et al.[15] made use of Indigo[47], PaDEL version 2.21 (ref. 48), RDKit (development version)[49] and the R Statistical Environment[50].

The atmospheric lifetimes were estimated with AOPWIN version 1.92a (ref. 51). The chemical structural analysis needed for additional corrections to the results of AOPWIN were performed with OpenBabel version 2.3.1 (ref. 52). As described in the main text, limited toxicity estimations were performed with the T.E.S.T. tool of the US Environmental Protection Agency[18].

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

## Acknowledgements

This work was supported by the US Department of Energy, Office of Energy Efficiency and Renewable Energy under contract no. DE-EE002057, with A. Bouza and B. Habibzadeh serving as Project Managers.

## Author contributions

P.A.D. and M.M. formulated the project, with M.M. serving as principal investigator. A.F.K. carried out the database screenings and associated estimation of properties. M.M. evaluated the stability and toxicity of the candidate molecules. J.S.B. and R.B. developed the cycle simulation model and performed the cycle simulations. P.A.D., R.B. and J.S.B. evaluated the cycle simulation results. M.M., P.A.D. and J.S.B. wrote the paper; and all authors contributed to revising the paper.

## Additional information

**Competing financial interests:** The authors declare no competing financial interests.

**Publisher's note**: 

