## [Peer Review File · Nature Communications]

Reviewers' comments:

Reviewer #1 (Remarks to the Author):

This is a good work to show the limited possible refrigerants to replace the current HFCs, the authors have screened thousands refrigerants based upon some critical parameters and criteria. At last a table of limited refrigerants was shown for references. This research results might be important for those policy makers and also research scientists for the limited options of environmental friendly refrigerants.

The whole paper has shown very clearly of the screen methods employed.

This manuscript could be accepted after minor revision.

The listed table should be linked with the results and conclusion part, thus let readers to understand the limited options of refrigerants.

It would be also necessary to indicate the potential applications groups for room air conditioning, centralized AC, and also refrigerators, and automotive AC.

Reviewer #2 (Remarks to the Author):

Review of NCOMMS-16-15589-T
McLinden et al. paper: Limited options.....

Summary of key results

The key results are clearly given in the last paragraph, and worded in such a way that the community involved in this issue is given a straightforward way to pursue. In fact, the "red line" presented in the paper going from a description of what is needed, to the method, and the conclusions is OK. There are a number of issues which would be helpful in the text, which will be given below.

Originality

The paper is original. It builds on further publications by McLinden, which have already certain elements presented in this paper again, but considered fully acceptable.

Data and methodology

The data analysed and the way it is done is good, gives clear boundaries before presenting results.

The methodology used is to analyse cycles with a large number of potential refrigerants and compare them against R-410A, the refrigerant used is AC in virtually all developed countries. However, here are a few points to mention. The various refrigerants are compared using the cycle standardised by ANSI(AHRI) given in line 114 of the PDF document. It would be good to describe which cycle that is (temperatures etc.) in the body of the document.

The paper compares all potential molecules to be used as refrigerants for the reference cycle using R-410A. This applies to the usage of R-410A in developed countries, where in developing countries, lots of AC equipment use HCFC-22 (R-22), and one would like to see how new molecules would compare to R-22. A bit more than a two line comment how R-22 and potential new molecules would compare (in relation to the current R-410A comparison) would be very helpful. In Figure 1, R-22 is already presented as the only ODP, high GWP fluid (molecule). A line in the concluding remarks addressing this would also be helpful.

In international talks on what to apply as a low GWP refrigerant in AC in future one often refers to

a good behaviour under high ambient temperature conditions. This would mean that a consideration of a different cycle than the ANSI/AHRI would be required. It may be possible to add a paragraph that the analysis applies to R-410A using standard cycles, plus what it would mean if different outside temperature conditions would be applied and whether that would have any consequences for the analysis presented (the screening).

Statistics

No problem, very much OK, also where it concerns the derivation of GWP data.

Conclusion

As mentioned, the conclusions in lines 218-230 are OK. It should be possible to add a few lines on comparison to R-22 and other temperature cycles (high ambient, i.e., higher condensation temperatures). That would make the paper even more valid in currently ongoing discussions.

Where it concerns Fig 1 and table 1, there is nothing from my side that should be considered for improvement. However, if the use of other molecules compared to R-22 would be highlighted, it might need some addition which is difficult for me to propose. I think that would be all manageable within the current framework.

Improvements overall and in details

The abstract is OK, may add toxicity in line 10, suitability in line 18. Line 35 mentions HCFC-22 (R-22, consistent approach in prefixes needed) but should mention that this refrigerant is next to R-410A is still commonly used in many (developing) countries, who have to convert to low GWP refrigerants. When line 142 mentions "optimised for the refrigerant" it may include a reference to not only R-410A, but also R-22. In the paper, slightly and mildly flammable refrigerants are defined; this is a bit confusing and one tries to go away from this. It would be good to just mention lower flammable refrigerants (class A2L). In line 216 it is not clear why R-134 is mentioned as a possible blend component. In principle the paper screens pure fluids (molecules) if one goes to blends of certain molecules it would need a bit more of an introduction somewhere, why and how.

References

Nothing to add

Clarity and context

The paper is clear, it works towards a clear outcome. If the above proposals on R-22 and a different high temperature cycle could be inserted, it would need to remain clear, i.e., only a small amount of comments would be needed in the conclusion. This could also be added in the abstract, if possible.

The methods paper (background) is good.

Editor and reviewer comments are given in Courier font.

Author replies are given in italicized Times font, and changes in the manuscript are highlighted.

Reviewer #1 (Remarks to the Author):

This is a good work to show the limited possible refrigerants to replace the current HFCs, the authors have screened thousands refrigerants based upon some critical parameters and criteria. At last a table of limited refrigerants was shown for references. This research results might be important for those policy makers and also research scientists for the limited options of environmental friendly refrigerants.

The whole paper has shown very clearly of the screen methods employed.

This manuscript could be accepted after minor revision.

Thank you for these comments.

The listed table should be linked with the results and conclusion part, thus let readers to understand the limited options of refrigerants.

We have added a specific reference to Table 1 ("the list") in the Discussion (page 9), so that readers skipping directly there understand the meaning of "list of best low-GWP fluids."

It would be also necessary to indicate the potential applications groups for room air conditioning, centralized AC, and also refrigerators, and automotive AC.

A statement regarding the more general applicability of our conclusions has been added to the Discussion (page 9).

Reviewer #2 (Remarks to the Author):

Review of NCOMMS-16-15589-T
McLinden et al. paper: Limited options.....

Summary of key results

The key results are clearly given in the last paragraph, and worded in such a way that the community involved in this issue is given a straightforward way to pursue. In fact, the "red line" presented in the paper going from a description of what is needed, to the method, and the conclusions is OK. There are a number of issues which would be helpful in the text, which will be given below.

Originality

The paper is original. It builds on further publications by McLinden, which have already certain elements presented in this paper again, but considered fully acceptable.

We do briefly summarize and reference our earlier, interim works. This is clarified with two new sentences added to the beginning of "Results." We feel that it is important to briefly summarize our earlier results to put the present results into context.

Data and methodology

The data analysed and the way it is done is good, gives clear boundaries before presenting results.

The methodology used is to analyse cycles with a large number of potential refrigerants and compare them against R-410A, the refrigerant used is AC in virtually all developed countries. However, here are a few points to mention. The various refrigerants are compared using the cycle standardised by

ANSI(AHRI) given in line 114 of the PDF document. It would be good to describe which cycle that is (temperatures etc.) in the body of the document.

Thank you for pointing out this oversight. The evaporator and condenser temperatures are now stated (page 5).

The paper compares all potential molecules to be used as refrigerants for the reference cycle using R-410A. This applies to the usage of R-410A in developed countries, where in developing countries, lots of AC equipment use HCFC-22 (R-22), and one would like to see how new molecules would compare to R-22. A bit more than a two line comment how R-22 and potential new molecules would compare (in relation to the current R-410A comparison) would be very helpful. In Figure 1, R-22 is already presented as the only ODP, high GWP fluid (molecule). A line in the concluding remarks addressing this would also be helpful.

This is a good point, and R-22 has been addressed in several ways in the revision:

page 2: It is explicitly stated that R-22 is still commonly used in developing countries.

page 6: The "cutoff" for the capacity is stated relative to R-22, in addition to R-410A.

page 7: We provide factors to "convert" the values in figures and tables from a R-410A to a R-22 reference.

page 9: A sentence has been added to the Discussion regarding the applicability of our conclusions for R-22 replacements (in addition to R-410A replacements).

In international talks on what to apply as a low GWP refrigerant in AC in future one often refers to a good behaviour under high ambient temperature conditions. This would mean that a consideration of a different cycle than the ANSI/AHRI would be required. It may be possible to add a paragraph that the analysis applies to R-410A using standard cycles, plus what it would mean if different outside temperature conditions would be applied and whether that would have any consequences for the analysis presented (the screening).

Our study focused on ambient conditions representative of North America, Europe, and Japan. Per reviewer's comment, we added a paragraph in the Results section (page 7) with an explanation how the screening results apply to higher ambient temperatures. A short sentence on this has also been added to the Discussion section (page 10).

Statistics

No problem, very much OK, also where it concerns the derivation of GWP data.

Conclusion

As mentioned, the conclusions in lines 218-230 are OK. It should be possible to add a few lines on comparison to R-22 and other temperature cycles (high ambient, i.e., higher condensation temperatures). That would make the paper even more valid in currently ongoing discussions.

A comparison to R-22 has been included in the Discussion section (page 9-10), as indicated above.

A short sentence on high-ambient conditions has been added to the Discussion section (page 10). Also see above for our detailed response to systems operating at high-ambient temperatures.

Where it concerns Fig 1 and table 1, there is nothing from my side that should be considered for improvement. However, if the use of other molecules compared to R-22 would be highlighted, it might need some addition which is difficult for me to propose. I think that would be all manageable within the current framework.

R-22 has been labeled in Figure 2 (which was Fig 1 in the original draft); this, together with the additional discussion on R-22 as outlined above, should make it easier for a reader more familiar with R-22 equipment to interpret our results.

Improvements overall and in details

The abstract is OK, may add toxicity in line 10, suitability in line 18. Line 35 mentions HCFC-22 (R-22, consistent approach in prefixes needed) but should mention that this refrigerant is next to R-410A is still commonly used in many

(developing) countries, who have to convert to low GWP refrigerants. When line 142 mentions "optimised for the refrigerant" it may include a reference to not only R-410A, but also R-22. In the paper, slightly and mildly flammable refrigerants are defined; this is a bit confusing and one tries to go away from this. It would be good to just mention lower flammable refrigerants (class A2L). In line 216 it is not clear why R-134 is mentioned as a possible blend component. In principle the paper screens pure fluids (molecules) if one goes to blends of certain molecules it would need a bit more of an introduction somewhere, why and how.

The reference to "safety properties" would include toxicity (as well as flammability and reactivity). The limit for the Abstract of 150 words does not allow elaboration of this point.

We have replaced the reference to "HCFC-22" and now consistently use the "R-xx" convention in all cases. See above for changes made to address R-22.

Unfortunately the classification scheme for flammability of refrigerants is more than "a bit confusing." That is the current state of the ASHRAE and ISO standards. We have attempted to summarize this scheme in a few words (page 8).

A short paragraph has been added to the Introduction (page 2-3) to introduce the concept of refrigerant blends and provide the rationale for our inclusion of several fluids (such as R-134) that would not be suitable as low-GWP refrigerants in their own right, but that might be useful as a blend component.

References

Nothing to add

Clarity and context

The paper is clear, it works towards a clear outcome. If the above proposals on R-22 and a different high temperature cycle could be inserted, it would need to remain clear, i.e., only a small amount of comments would be needed in the conclusion. This could also be added in the abstract, if possible.

As presented above, we feel that we have addressed the reviewers comments on R-22 and high-temperature cycles. Unfortunately, space limitations do not allow inclusion of these points in the Abstract.

The methods paper (background) is good.

REVIEWERS' COMMENTS:

Reviewer #1 (Remarks to the Author):

I am satisfied with the replies from the authors to my comments and suggestions, now It can be accepted as is.

Reviewer #2 (Remarks to the Author):

Going through the revised version (merged document), I think that all of my comments have been adequately responded to. I also support other information that has been added. I have no further review comments.